# Isoorientin Inhibits Amyloid β_25–35_-Induced Neuronal Inflammation in BV2 Cells by Blocking the NF-κB Signaling Pathway

**DOI:** 10.3390/molecules26227056

**Published:** 2021-11-22

**Authors:** Buyun Kim, Ki Yong Lee, Byoungduck Park

**Affiliations:** 1College of Pharmacy, Keimyung University, 1095 Dalgubeoldaero, Dalseo-gu, Daegu 704-701, Korea; bykim@kiom.re.kr; 2Korean Medicine (KM) Application Center, Korea Institute of Oriental Medicine (KIOM), Dong-gu, Daegu 701-300, Korea; 3College of Pharmacy, Korea University, Sejong Campus, 2511 Sejong-ro, Sejong 339-770, Korea

**Keywords:** Alzheimer’s disease, amyloid-beta, ROS, NF-κB, inflammation

## Abstract

Alzheimer’s disease (AD) is a severe neurodegenerative disorder. AD is pathologically characterized by the formation of intracellular neurofibrillary tangles, and extracellular amyloid plaques which were comprised of amyloid-beta (Aβ) peptides. Aβ induces neurodegeneration by activating microglia, which triggers neurotoxicity by releasing various inflammatory mediators and reactive oxygen species (ROS). Nuclear factor-kappa B (NF-κB) is expressed in human tissues including the brain and plays an important role in Aβ-mediated neuronal inflammation. Thus, the identification of molecules that inhibit the NF-κB pathway is considered an attractive strategy for the treatment and prevention of AD. Isoorientin (3′,4′,5,7-Tetrahydroxy-6-C-glucopyranosyl flavone; ISO), which can be extracted from several plant species, such as *Philostachys* and *Patrinia* is known to have various pharmacological activities such as anticancer, antioxidant, and antibacterial activity. However, the effect of ISO on Aβ-mediated inflammation and apoptosis in the brain has yet to be elucidated. In the present study, we investigated whether ISO regulated Aβ-induced neuroinflammation in microglial cells and further explored the underlying mechanisms. Our results showed that ISO inhibited the expression of iNOS and COX-2 induced by Aβ_25–35._ And, it inhibited the secretion of pro-inflammatory cytokines such as tumor necrosis factor-α (TNF-α) and interleukin-6 (IL-6). In addition, ISO reduced the ROS production in Aβ_25–35_-induced BV2 cells and inhibited NF-κB activation. Furthermore, ISO blocked Aβ_25–35_-induced apoptosis of BV2 cells. Based on these findings, we suggest that ISO represents a promising therapeutic drug candidate for the treatment and prevention of AD.

## 1. Introduction

Alzheimer’s disease (AD) is a neurodegenerative disorder affecting memory, language, and cognitive function. [1]. The pathology of AD is characterized by the presence of senile plaques, neurofibrillary tangles, and neuronal loss in the brain [2]. In particular, amyloid-beta (Aβ), which is known to play an important role in the development of AD, is deposited in the form of senile plaques with various molecular properties [3]. Aβ induces neurodegeneration by activating microglia, which triggers neurocytotoxicity by releasing inflammatory mediators such as cytokines, nitric oxide (NO), and reactive oxygen species (ROS) resulting in AD progression [4,5,6]. ROS activate the expression of protein kinase C (PKC) and mitogen-activated protein kinase (MAPK) in the downstream signal transduction pathway resulting in nuclear translocation of nuclear factor kappa B (NF-κB) and the expression of pro-inflammatory genes [7]. NF-κB, a transcription factor that is expressed in human tissues including the brain, plays an important role in Aβ-mediated neuronal inflammation. Activated NF-κB complex in the cytoplasm migrates to the nucleus and binds to DNA consensus sequences to promote the expression of genes related to inflammation [8]. Thus, drugs that inhibit the NF-κB pathway represent an attractive treatment and prevention strategy against AD by modulating various target molecules related to oxidative stress, apoptosis, and inflammation.

Isoorientin (3′,4′,5,7-Tetrahydroxy-6-C-glucopyranosyl flavone; ISO), a common C-glycosylflavone, can be extracted from several plant species, such as *Philostachys* and *Patrinia* [9,10]. Plant extracts containing ISO exhibit anticancer, antioxidant, anti-bacterial, and anti-nociceptive activities [11,12,13,14]. However, the effect of ISO on Aβ-mediated inflammatory progression and apoptosis in the brain region has yet to be elucidated. Since BV2 cells are activated by oxidative stress or inflammatory factors, which can possibly lead to neurodegenerative disorders such as Alzheimer’s and Parkinson’s disease, BV2 cells are widely used as an alternative model for primary microglia and neurodegenerative disease models in vitro. Therefore, we investigated whether ISO regulates Aβ-induced neuroinflammation using BV2 cells and further explored the underlying mechanisms.

## 2. Results

### 2.1. ISO Reverses the Cytotoxic Effects of Aβ_25–35_ in BV2 Microglial Cells

The amyloid-beta peptide is commonly found in Alzheimer’s disease (AD). Particularly, the Aβ_25–35_ fragment has been known to play a pivotal role in AD, due to its peculiar aggregation properties [11]. Based on a previous study reporting inflammation in the microglial BV2 cells upon treatment with 20 μM Aβ_25–35_, we used this concentration throughout this study. First, we investigated whether ISO prevented the cytotoxicity induced by Aβ_25–35_. To determine the optimal concentration of ISO, we evaluated the cell viability at the indicated concentrations of ISO in the absence or presence of 20 μM Aβ_25–35_ using CCK-8 assays. The results showed that 20 μM ISO restored cell viability to control levels in BV2 cells (Figure 1B). These results suggest that ISO can reverse the cytotoxic effects of Aβ_25–35_.

### 2.2. ISO Inhibits the Expression of iNOS and COX-2 Induced by Aβ_25–35_

Aβ-induced microglial activation promotes chronic inflammation and neuronal cell death by releasing various molecules such as NO, prostaglandin E_2_ (PGE_2_), inducible nitric oxide synthase (iNOS), and cyclooxygenase-2 (COX-2) [12,13]. Since iNOS and COX-2 are pivotal enzymes for the production of NO and PGE2, we analyzed their expression at the transcriptional and translational level in Aβ_25–35_-stimulated BV-2 cells. The results demonstrated that 20 μM ISO inhibited the upregulation of iNOS and COX-2 induced by Aβ_25–35_ both at the mRNA and protein levels (Figure 1C,D). We further investigated whether ISO suppressed the NO production in Aβ-induced BV2 cells. Our results showed that Aβ-induced NO production was inhibited by ISO (Figure 1E).

### 2.3. ISO Suppresses Aβ_25–35_-Induced ROS Generation and Expression of TNF-α and IL-6 in BV2 Cells

ROS synthesis by Aβ in the microglia contributes to oxidative neuronal damage and neurodegeneration, resulting in neurological diseases [14,15]. Therefore, we investigated whether the anti-inflammatory effect of ISO was mediated by decreased ROS production. As shown in Figure 2A, 20 μM Aβ_25–35_ increased ROS synthesis; however, pretreatment with ISO significantly reduced ROS levels in a dose-dependent manner. These data suggest that ISO inhibits inflammatory progression by ameliorating ROS generation in BV2 cells.

Accumulation and aggregation of Aβ peptides generally lead to activation of neuroglia cells, which eventually initiates neuronal oxygen response and release of inflammatory cytokines such as interleukin-1β (IL-1β), IL-6, and tumor necrosis factor-α (TNF-α) [16,17]. We investigated whether ISO inhibited the release of inflammatory cytokines after Aβ_25–35_ treatment. As shown in Figure 2B,C, the levels of TNF-α and IL-6 were increased in Aβ_25–35_-induced BV2 cells. However, pretreatment with ISO significantly reduced the synthesis of these cytokines both at the protein and mRNA levels.

### 2.4. ISO Inhibits Aβ_25–35_-Mediated NF-κB Signaling Pathway

The NF-κB pathway is well known to promote the expression of genes related to neuronal inflammation [18,19,20]. Our cumulative results demonstrated the anti-inflammatory effects of ISO. Therefore, we determined whether the effect of ISO was linked to inhibition of the NF-κB pathway in BV2 cells. As expected, the Aβ_25–35_ treatment increased the phosphorylation of IκB protein, whereas ISO inhibited this activation (Figure 3A). In addition, ISO abrogated the NF-κB-DNA binding activity induced by Aβ_25–35_ (Figure 3B). Next, we investigated whether ISO affected the nuclear translocation of the NF-κB complex. As shown in Figure 3C, ISO inhibited the nuclear translocation of NF-κB in Aβ_25–35_-stimulated BV2 cells. These results indicated that ISO inhibited the inflammatory process in Aβ_25–35_-induced BV2 cells via blockade of the NF-κB pathway.

### 2.5. ISO Blocks Aβ_25–35_-Induced Apoptosis in BV2 Microglial Cells

Aβ accelerates neurodegeneration and promotes neuronal cell apoptosis in AD patients [21]. Besides, Aβ plaques induce cellular apoptosis by regulating mitochondrial depolarization, which induces oxidative stress [22]. Therefore, we investigated whether ISO affected the expression of various proteins involved in apoptotic progression. As shown in Figure 4A, the level of anti-apoptotic protein Bcl-2 was decreased, while the level of pro-apoptotic protein BAX was increased upon treatment of BV2 cells with 20 μM Aβ_25–35._ However, ISO reversed the expression of Bcl-2 and BAX. We then analyzed the expression of cleaved caspases-9 and -3 as well as PARP, which are markers of apoptosis. Aβ promoted the cleavage of these proteins, whereas ISO treatment abrogated these effects (Figure 4B). These results suggested that ISO has an inhibitory effect on neuronal cell apoptosis induced by Aβ_25–35_.

## 3. Discussion

The precise mechanism of AD is still not known; however, many studies revealed that neuronal inflammation is closely related to the development of AD. The process of neuroinflammation is marked by the production of pro-inflammatory cytokines, including IL-1β, IL-6, tumor necrosis factor (TNF), NO, and ROS. The innate immune cells involved in this process are microglia and astrocytes [23]. Particularly, inflammation in the brain region is mainly mediated by glial cells [12]. Microglia activated by Aβ produce a wide range of pro-inflammatory cytokines, leading to nerve damage and cell death [16]. Thus, a drug that inhibits Aβ-induced microglial activation represents a promising therapeutic candidate targeting neurodegeneration in AD. The purpose of this study was to investigate the protective effects of ISO against neuro-inflammatory damage induced by Aβ_25–35_ in BV2 microglial cells and to analyze the underlying mechanism accordingly.

Once the microglial cells are activated by Aβ, an unregulated neuro-inflammatory response, i.e., an excessive release of pro-inflammatory molecules such as cytokines, chemokines, NO, and ROS, induces neurodegeneration. Therefore, the identification of molecules that inhibit the release of these inflammatory substances is a robust strategy to treat neurodegenerative diseases [24]. Initially, we investigated whether ISO, a common C-glycosylflavone, affected the expression of molecules involved in neuroinflammation. The results showed that ISO inhibited the expression of various neuro-inflammatory molecules such as iNOS, COX-2, TNF-α, and IL-6 in Aβ_25–35_-induced BV2 cells. We elucidated the mechanism involved. ROS are well known to amplify neuronal inflammatory signals in microglia via activation of MAPKs, and transcription factors such as NF-κB as well as overexpression of neuro-inflammatory molecules [25,26,27]. Another study reported that ROS accumulation was strongly associated with microglial inflammation in neurodegenerative diseases [28]. Thus, we measured the ROS generation during Aβ_25–35_-induced neuroinflammation in BV2 cells. As shown in Figure 2A, Aβ_25–35_ increased the ROS production; however, pretreatment with ISO significantly reduced ROS levels in a dose-dependent manner. These data suggest that ISO inhibits the progression of neuroinflammation by attenuating ROS generation in BV2 cells.

It has been reported that infusion of Aβ into the brain may induce cognitive damage and nerve inflammation by promoting nuclear translocation of NF-κB and activating the MAPK signaling pathway [29]. NF-κB signaling is a representative pathway in the regulation of inflammation. This pathway is activated via phosphorylation of upstream IKK, followed by cleavage of IκB, an inhibitory protein of NF-κB, and separation of NF-κB complex. Activated NF-κB is translocated to the nucleus to regulate the expression of various genes involved in inflammation [30]. In the present study, Aβ_25–35_ promoted the phosphorylation of IκB in the cytoplasm, which was abrogated by ISO. Moreover, ISO suppressed the NF-κB-DNA binding activity in the nucleus. These results implied that ISO alleviates neuronal inflammation via regulation of NF-κB activation.

Apoptosis, also known as programmed cell death, is a basic biological process underlying several important functions in developmental organisms [31]. However, abnormal induction of apoptosis, especially in neuronal cells, can lead to serious consequences. Apoptosis, which is abnormally induced in the brain region, has been shown to contribute to the development and progression of AD [32]. Besides, microglial activation promotes the synthesis of inflammatory cytokines leading to neuronal cell death [21]. Another study found that Aβ peptides induce mitochondria-mediated apoptotic pathways [22]. Similarly, our results showed that Aβ_25–35_ promotes the expression of pro-apoptotic protein BAX and decreases the levels of anti-apoptotic protein Bcl-2 in BV2 microglial cells. Conversely, ISO restored the expression of these proteins in microglial cells. Furthermore, ISO blocked the cleavage of caspase-9, -3, and PARP.

Overall, the results showed that ISO ameliorated neuronal inflammation via inhibition of ROS generation and blockade of NF-kB activity. ISO also showed a protective effect on neuronal cell apoptosis induced by Aβ_25–35_. Therefore, we propose that ISO represents a promising therapeutic drug candidate for the treatment and prevention of AD (Figure 5).

## 4. Materials and Methods

### 4.1. Purity Analysis of Isoorientin

Isoorientin (ISO) is a flavonoid isolated from Polygonum orientale [33]. Purity analysis of isolated ISO was performed on a Waters 2695 system coupled with a photodiode array detector. The chromatographic separation was carried out on a Shiseido CapCell PAK C18 column (4.6 mm I.D × 150 mm, 5 μm) using 0.1% formic acid (solvent A) and acetonitrile with 0.1 % formic acid (solvent B). The gradient elution was followed as 5% of B at 0–5 min, 5–95% of B at 5–30 min. The flow rate was 0.6 mL/min and the injection volume was 10 µL. The column was thermostatted at 25 °C. The UV chromatogram was monitored at 330 nm. The purity of isoorientin was 98.1 ± 0.3%.

### 4.2. Cell Culture

The murine BV2 microglial cells were obtained from the American Type Culture Collection (ATCC, Manassas, VA, USA) and cultured in Dulbecco’s Modified Eagle Medium (DMEM) supplemented with 10% fetal bovine serum (FBS) and 1% antibiotic-antimycotic at 37 °C in a humidified incubator (5% CO_2_, 95% air). DMEM, FBS, 0.25% Trypsin-EDTA, and antibiotic-antimycotic were obtained from Gibco (Grand Island, NY, USA).

### 4.3. Cell Cytotoxicity Assay

Cell cytotoxicity was determined by a quantitative colorimetric assay using WST-8 (DOJINDO Laboratories, Kumamoto, Japan). Aβ peptide fragment 25–35 (Aβ_25–35_) was purchased from Sigma Chemical Co (St. Louis, MO, USA). BV2 cells were seeded at a density of 1 × 10^4^ cells/well in 96 well polystyrene culture plates at 37 °C with 5% (*v*/*v*) CO_2_. Subsequently, various concentrations of ISO and 20 μM Aβ_25–35_ were added to BV2 cells and incubated for 24 h. Then, 10 μL of WST-8 reagent was added to each well according to the manufacturer’s instructions. The plate was incubated for 4 h at 37 °C and measured at an absorbance of 450 nm by a plate reader (Tecan Sunrise, Tecan Group AG, Zürich, Switzerland).

### 4.4. Western Blotting Analysis

BV2 cells were washed three times with PBS and lysed for 10 min using a RIPA buffer (150 mM NaCl, 1% NP-40, 0.5% sodium deoxycholate, 0.1% SDS, 50 mM Tris-HCl (pH 7.4), 50 mM glycerophosphate, 20 mM NaF, 20 mM EGTA, 1 mM DTT, 1 mM Na_3_VO_4_, and protease inhibitors) at 4 °C. After centrifugation at 15,000 rpm for 10 min, the supernatant was separated. In brief, 30 μg proteins were separated by SDS-PAGE and transferred to the PVDF membrane (Invitrogen, Carlsbad, CA, USA). After being blocked with 5% nonfat dry milk, the membrane was incubated at 4 °C overnight with the indicated antibodies: iNOS (1:2000, ab3523, Abcam, Cambridge, UK), COX-2 (1:1000, #12282, Cell signaling, Beverly, MA, USA), TNF-α (1:1000, #8902, Cell signaling), IL-6 (1:1000, #12153, Cell signaling), phospho-IκBα (1:1000, #2859, Cell signaling), IκBα (1:1000, #4812, Cell signaling), BAX (1:1000, sc-493, Santa Cruz Biotechnology, Santa Cruz, CA, USA), Bcl-2 (1:1000, sc-492, Santa Cruz), cleaved caspase-3 (1:1000, #9661, Cell signaling), cleaved caspase-9 (1:1000, #7237, Cell signaling), cleaved PARP (1:1000, #5625, Cell signaling) and β-Actin (1:1,000, #4970, Cell Signaling). After incubation with primary antibodies, each membrane was incubated with an appropriate dilution of HRP-conjugated anti-rabbit IgG (1:5000, w4018, Promega, Madison, WI, USA) and anti-mouse IgG (1:5000, w4028, Promega) for 1 h. The immune complexes were visualized with the ECL system (Amersham Pharmacia Biotech Inc, Arlington Heights, IL, USA), and the bands were quantified by Fusion Solo software (Vilber Lourmat, Marne-la-Vallée, France).

### 4.5. Reverse Transcription Polymerase Chain Reaction (RT-PCR)

Total RNA was isolated from BV2 cells using TRIZOL reagent (Invitrogen Co, Grand Island, NT, USA) according to the manufacturer’s instruction. cDNA was synthesized by reverse transcription from 1 μg of total RNA using *AccuPower* Rocketscript cycle RT premix (Bioneer, Daejeon, Korea). Aliquots of cDNA were used for PCR using primer sets specific to iNOS, COX-2, IL-6, TNF-α, and GAPDH as a control. Used primers are as follows: iNOS sense: 5′-AGACCTCAACAGAGCCCTCA-3′, antisense: 5′-GCAGCCTCTTGTCTTTGACC-3′; COX-2 sense: 5′-GGAGAGACTATCAAGATAGT-3′, antisense: 5′-ATGGTCAGTAGACTTTTACA-3′; IL-6 sense: 5′-CCGGAGAGGAGACTTCACAG-3′, antisense: 5′-TCCACGATTTCCCAG-AGAAC-3′; TNF-α sense: 5′-TCAGCCTCTTCTCATTCCTG-3′, antisense: 5′-TGAAGAGAACCTGGGAGTAG-3′; GAPDH sense: 5′-TTGCAGTGGCAAAGTGGAGA-3′, antisense: 5′-CGTGGTTCACACCCATCACAA-3′.

### 4.6. Measurement of Intracellular ROS

Intracellular ROS levels were measured by the fluorescent probe 2,7-dichlorofluorescein diacetate (DCF-DA). BV2 cells were pretreated with or without indicated concentration of ISO for 1 h and added with 20 μM Aβ_25–35_ for 24 h. Then, BV2 cells were labeled with 25 μM DCF-DA for 30 min at 37 °C in a 5% CO_2_ incubator. After washing the cells twice with PBS, 1ml PBS was added and intracellular ROS was measured by fluorescence microscopy (Nikon Instruments Inc., New York, NY, USA).

### 4.7. Measurement of Nitrite Production

The nitrite accumulated in the culture medium was measured as an indicator of NO production using the Griess reagent kit (Promega, Madison, WI, USA). Cell culture media were centrifuged at 4 °C for 5 min and the supernatant was subjected to measurement of nitrite.

### 4.8. Electrophoretic Mobility Shift Assay (EMSA)

Measurement of NF-κB-DNA binding activity in BV2 cells was applied to EMSA analysis as described in previous studies [34]. The nuclear protein was harvested and NF-κB-DNA binding activity was measured according to the manufacturer’s instructions.

### 4.9. Immunofluorescence Assay

BV2 cells (5 × 10^5^ cells) were seeded on sterile coverslips and pretreated with 20 μM ISO for 1 h and added with 20 μM Aβ_25–35_ for 24 h. After washing with PBS three times, the cells were fixed in 4% paraformaldehyde for 10 min at room temperature. After washing with PBS three times, cells on coverslips were permeabilized in 0.2% (*v*/*v*) Triton X-100 for 5 min at room temperature. Cells were washed again with PBS, the coverslips were blocked with PBS containing 5% BSA for 5 min and incubated with an antibody against p-NF-κB (sc-166748, Santa Cruz Biotechnology, Santa Cruz, CA, USA) diluted with PBS containing 5% BSA (1:500) for 1 h at room temperature. After washing with PBS three times, the coverslips were incubated with a secondary antibody Alexa Flour 488 (1:500, Invitrogen, Carlsbad, CA, USA) for 30 min and counterstained for nuclei with Hoechst 33,342 for 10 min. After washing with PBS three times, the coverslips were mounted using Prolong Gold antifade Mountant reagent (Life Technologies, Carlsbad, CA, USA). Each coverslip was analyzed on the Nikon Eclipse Ti fluorescence microscope (Nikon Instruments Inc., New York, NY, USA).

### 4.10. Statistical Analysis

The data were presented as Mean ± SEM and analyzed using Graph Pad Prism 6.0 (GraphPad Software, La Jolla, CA, USA). Statistical significances among treatment groups were compared using the one-way analysis of variance (ANOVA) with repeated measures (RM), followed by Bonferroni’s post hoc test, respectively. A significant difference was indicated by p-values less than 0.05, 0.01, or 0.001.

## Figures and Tables

**Figure 1 molecules-26-07056-f001:**
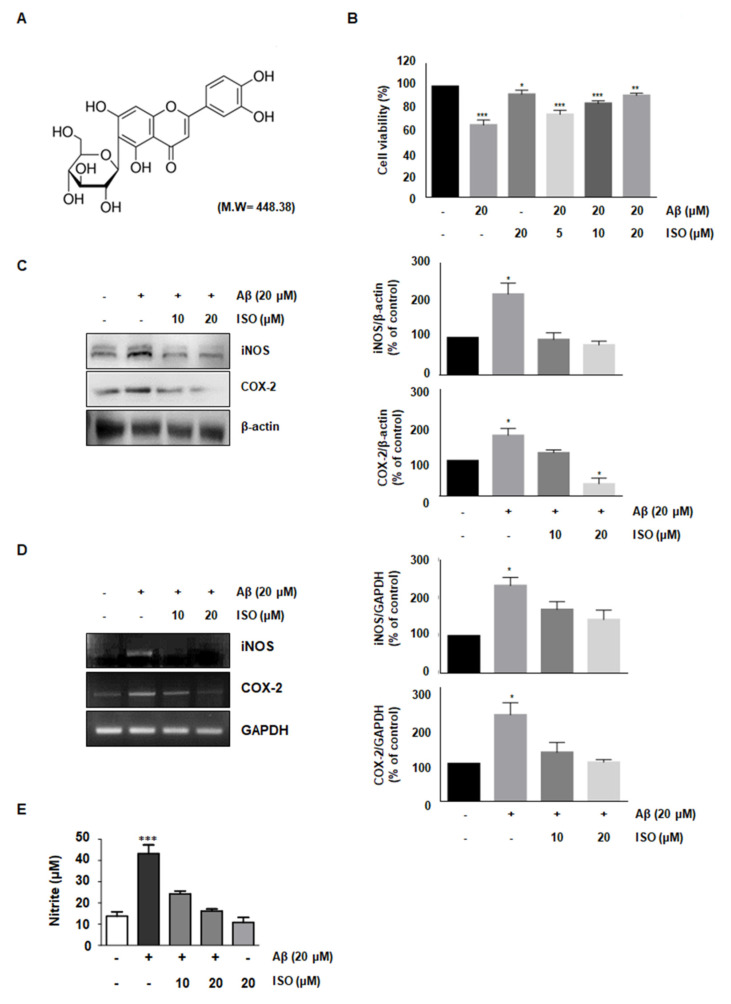
ISO reverses the cytotoxic effects of Aβ_25–35_ in BV2 cells. (**A**) Structure of isoorientin (ISO) (**B**) BV2 cells (1 × 10^4^ cells/mL) were treated with the indicated concentrations of ISO (0, 5, 10, 20 μM) 1 h before Aβ_25–35_ (20 μM) treatment for 24 h. Cell viability was assessed by CCK-8 assays and the results are expressed as a percentage of surviving cells over control cells. Results are representative of those obtained from three independent experiments. (**C**) BV2 cells were pretreated with different concentrations of ISO as indicated 1 h before the addition of Aβ_25–35_ (20 μM) for 24 h. The protein levels were observed by Western blotting analysis. (**D**) The mRNA levels of iNOS and COX-2 were determined by the RT-PCR in BV2 cells. (**E**) BV2 cells were pretreated with different concentrations of ISO as indicated 1 h before the addition of Aβ_25–35_ (20 μM) for 24 h. Cell culture media were harvested for the measurement of nitrite (NO). The experiments were repeated more than three times and similar results were obtained. Data indicate mean ± SEM of three independent experiments. * *p* < 0.05, ** *p* < 0.01, *** *p* < 0.001 versus control.

**Figure 2 molecules-26-07056-f002:**
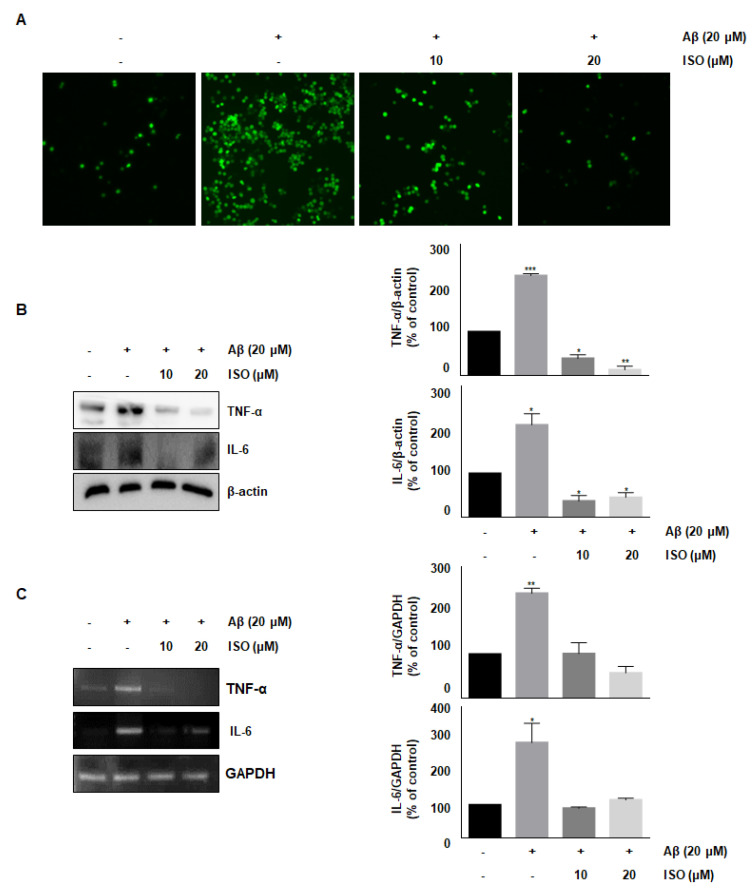
ISO inhibits Aβ_25–35_-induced ROS generation and expression of inflammatory cytokines in BV2 cells. BV2 cells were pretreated with different concentrations of ISO as indicated 1 h before the addition of Aβ_25–35._ (**A**) Intracellular ROS levels were measured by the DCF-DA. (**B**) The protein levels of TNF-α and IL-6 were determined by Western blotting. (**C**) The mRNA levels of TNF-α and IL-6 were determined by the RT-PCR. β-actin and GAPDH were used as loading controls. The experiments were repeated more than three times and similar results were obtained. Data indicate mean ± SEM of three independent experiments. * *p* < 0.05, ** *p* < 0.01, *** *p* < 0.001 versus control.

**Figure 3 molecules-26-07056-f003:**
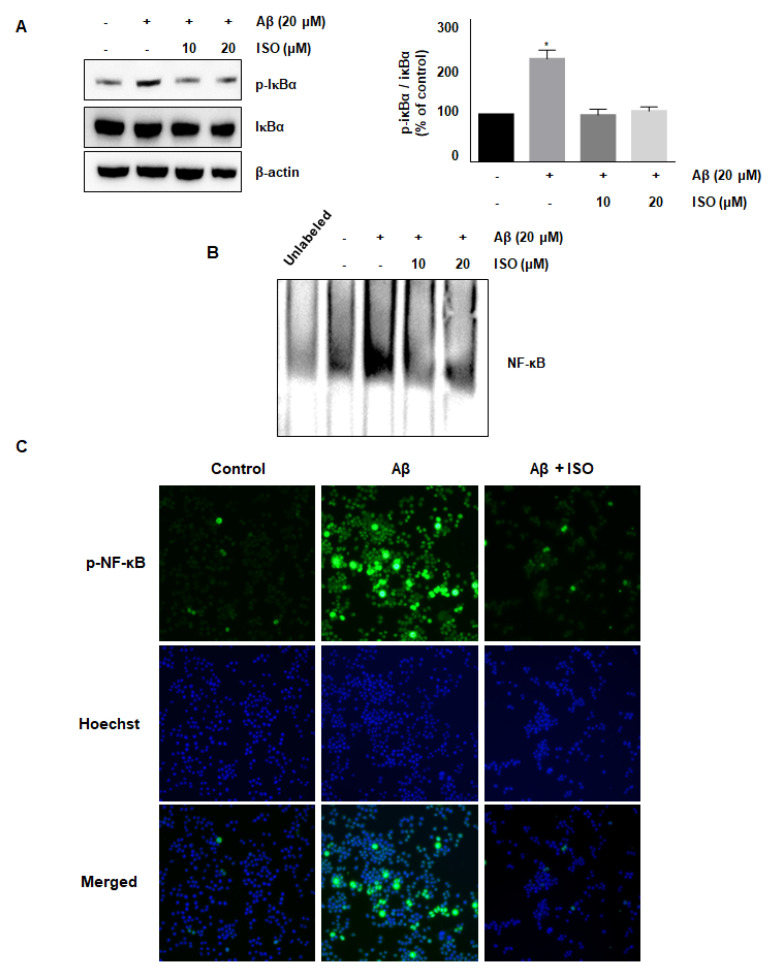
ISO inhibits the Aβ_25–35_-mediated NF-κB signaling pathway. BV2 cells were pretreated with different concentrations of ISO as indicated 1 h before the addition of Aβ_25–35._ (**A**) The phosphorylation level of IκBα was determined by Western blotting using a cytosolic extract. Data indicate mean ± SEM of three independent experiments. * *p* < 0.05 versus control (**B**) Nuclear extracts of BV2 cells were analyzed by EMSA. (**C**) The immunofluorescence assay was performed to detect NF-κB nuclear localization. Stained BV2 cells were visualized by a fluorescence microscope (200× magnification).

**Figure 4 molecules-26-07056-f004:**
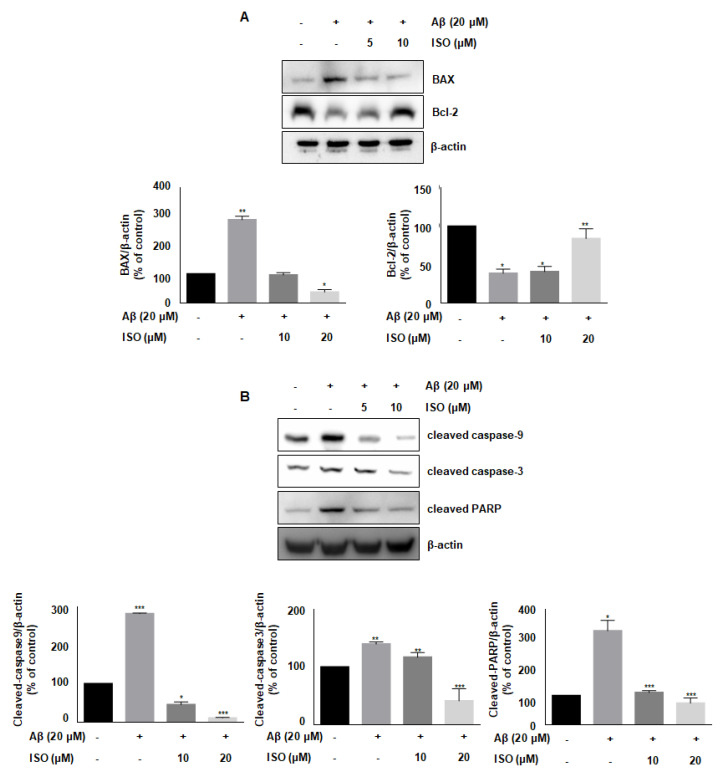
ISO blocks Aβ_25–35_-induced apoptosis in BV2 microglial cells. BV2 cells were pretreated with different concentrations of ISO as indicated 1 h before the addition of Aβ_25–35._ (**A**) The protein levels of Bax and Bcl-2 were observed by Western blot analysis. (**B**) The levels of cleaved caspase-9, -3, and PARP were observed by Western blot analysis. β-actin was used as loading controls. The experiments were repeated more than three times and similar results were obtained. Data indicate mean ± SEM of three independent experiments. * *p* < 0.05, ** *p* < 0.01, *** *p* < 0.001 versus control.

**Figure 5 molecules-26-07056-f005:**
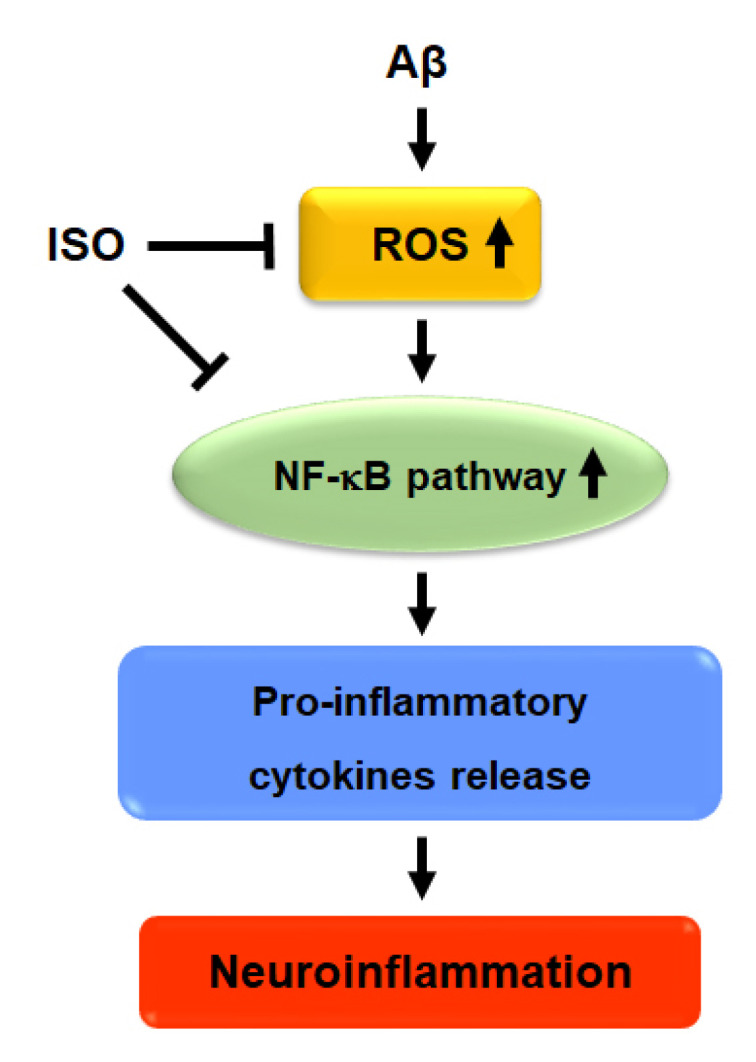
Proposed signaling mechanism of ISO on Aβ-induced neuroinflammation.

## Data Availability

Not available.

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
