# Peer review of "Isoorientin Inhibits Amyloid β25–35-Induced Neuronal Inflammation in BV2 Cells by Blocking the NF-κB Signaling Pathway"

_molecules, 2021, doi:10.3390/molecules26227056_

Round 1

Reviewer 1 Report

The manuscript by Kim et al., investigates the effects of isoorientin (3′,4′,5,7-Tetrahydroxy-6-C-glucopyranosyl flavone; ISO), a common C-50 glycosylflavone, extracted from several plant species, on Aβ-mediated inflammation and apoptosis in microglial BV2 cells. In addition, the Authors investigated the underlying signalling mechanisms. Specifically, the Authors found that ISO inhibited the expression of iNOS and COX-2 induced by Aβ25-35 peptide, inhibited the secretion of pro-inflammatory cytokines TNF-α and IL-6. Furthermore, ISO reduced the ROS production, inhibited NF-κB activation and blocked Aβ25-35-induced apoptosis of Aβ25-35-treated BV2 cells.

The manuscript is of interest in the field of neuroinflammation representing an interesting way to modulate inflammatory processes leading to neurodegeneration.

The manuscript should be improved addressing these indications:

  • To explain why BV2 cells were used instead of primary microglia cells
  • A schematic diagram reporting the results obtained should be added
  • Aβ peptide fragment 25–35 purchased from Sigma Chemical Co (St. Louis, MO). The Authors should better explain how the peptide was dissolved and whether it was tested as a monomer or as aggregated form. Did Aβ peptide form fibrils in cell culture medium?
  • The effects of ISO alone on iNOS, COX2, TNFalpha and IL6 should be reported. Why the combination of Aβ peptide with ISO reduced COX2, TNFalpha and IL6 levels under 100% of the control? The same for apoptotic factors.

Author Response

* For your convenience, we highlighted all figures and phrases modified in revised manuscript in blue.

< Reviewer 1 >

  1. To explain why BV2 cells were used instead of primary microglia cells

Answer

Thank you for your good advice. As your suggestion, we confirmed and revised. (Page 2; Line 54-59)

  1. A schematic diagram reporting the results obtained should be added

Answer

As your suggestion, we revised attached new figure 5 file.

  1. Aβ peptide fragment 25–35 purchased from Sigma Chemical Co (St. Louis, MO). The Authors should better explain how the peptide was dissolved and whether it was tested as a monomer or as aggregated form. Did Aβ peptide form fibrils in cell culture medium?

Answer

Thank you for your good advice. Sigma Co. suggests 1 % of acetic acid as a solvent for Aβ peptide, however, we dissolved it in 1X PBS after screening. Once it was well dissolved, we think PBS would be more suitable solvent than acetic acid due to its safety. We could not detect any aggregated form or fibrils in BV2 cells.

  1. The effects of ISO alone on iNOS, COX2, TNFalpha and IL6 should be reported. Why the combination of Aβ peptide with ISO reduced COX2, TNFalpha and IL6 levels under 100% of the control? The same for apoptotic factors.

Answer:

Previous study showed that there was no effect of ISO alone on iNOS, COX2, TNF-alpha and IL6 expression (Yuan et al., Isoorientin attenuates lipopolysaccharide-induced pro-inflammatory responses through down-regulation of ROS-related MAPK/NF-kappaB signaling pathway in BV-2 microglia. Mol Cell Biochem 2014, 386, (1-2), 153-65). Based on these findings, we focused on the anti-inflammatory and anti-apoptotic effects of ISO in Aβ-induced BV2 cells. We think the reason that ISO reduced the proteins involved in inflammation and apoptosis under the control group is considered as an inhibition of ROS/NF-kB pathway. Strong inhibition of inflammatory pathway by ISO decreased the protein expression even under basal level.

Reviewer 2 Report

This research was focused on the activities and mechanisms of ISO in Abeta-activated microglia in vitro. Also, the proinflammatory enzymes, mediators, and cytokines were evaluated. However, some issues should be clarified. And, English grammar needs to be improved.

  1. The source and purity of ISO and its vehicle (%) should be mentioned in the text.
  2. What is the vehicle of Abeta?
  3. The statistical significance should be labeled in Fig. 1.
  4. There were two bands of iNOS mRNA by RT-PCR (Fig. 1D), it needs to clarify the condition.
  5. The NO level should be evaluated at three concentrations of ISO.
  6. Appropriately, merge sections 3.3 and 3.4 for Fig. 2.
  7. The statistical analyses of Fig. 3C should be included.
  8. The statistical significance should be labeled in Fig. 4B (cleaved caspase-3/ PARP).
  9. The content of the discussion should be more expanded and focused on neuroinflammatory issues.
  10. The MAO of NFkB (IKK, TAK-1) should be in-depth discussed. 

Author Response

* For your convenience, we highlighted all figures and phrases modified in revised manuscript in blue.

< Reviewer 2 >

  1. The source and purity of ISO and its vehicle (%) should be mentioned in the text.

Answer

As your suggestion, we confirmed and revised. (Page 2; Line 61-69)

  1. What is the vehicle of Abeta?

Answer

We dissolved it in 1X PBS.

  1. The statistical significance should be labeled in Fig. 1.

Answer

As your suggestion, we revised. (Page 5; Line 176-177)

  1. There were two bands of iNOS mRNA by RT-PCR (Fig. 1D), it needs to clarify the condition.

Answer

As your suggestion, we revised and attached new figure 1 file.

  1. The NO level should be evaluated at three concentrations of ISO.

Answer

As your suggestion, we performed experiments again and inserted to figure 1. (Page 3; Line 125-129, Page 5; Line 173-175, Line 185-187)

  1. Appropriately, merge sections 3.3 and 3.4 for Fig. 2.

Answer

As your suggestion, we revised. (Page 6; Line 188-211)

  1. The statistical analyses of Fig. 3C should be included.

Answer

Thank you for your advice. In my opinion, since the data of figure 3C are visualized by fluorescence, it doesn’t need to be quantified. However, if you still think the statistical analyses are needed, I will try.

  1. The statistical significance should be labeled in Fig. 4B (cleaved caspase-3/ PARP).

Answer

As your suggestion, we revised and attached new figure 4 file.

  1. The content of the discussion should be more expanded and focused on neuroinflammatory issues.

Answer

As your suggestion, we revised. (Page 9; Line 252-255)

  1. The MAO of NFkB (IKK, TAK-1) should be in-depth discussed.

Answer

Thank you for your warm advice. In my opinion, we already described the process of NF-kB activation via IKK phosphorylation and downstream pathway. If you think that we further need to discuss on it, please give me the direction in detail, then I will try to discuss.

Round 2

Reviewer 1 Report

The manuscript has been improved, as requested, and now it is suitable for publication.